# Peer review of "Mode II Fatigue Delamination Growth and Healing of Bis-Maleimide Modified CFRPs by Using the Melt Electro-Writing Process Technique"

_jcs, doi:10.3390/jcs7090350_

Round 1

Reviewer 1 Report

###### GENERAL COMMENTS ######

# In this manuscript, Mode II fatigue delamination growth and healing of Bis-maleimide modified CFRPs were investigated using the MEP technique. The investigation focused on the evaluation of the influence of fatigue life estimation. Mode II fatigue loading conditions were used for delamination growth of CFRPs modified with BMI resin (containing or not containing GNPs). Quasi-static Mode II and Mode II fatigue experimental tests were carried out. The authors found a betterd delamination behavior after healing activation.

# Interesting subject. Interesting research.

#The research methodology raises some doubts that I would like to see clarified:

-- How valid are the fatigue tests? I think the load cell capacity is too large for the values read.

-- the validation of the correlation curves. Pearson's correlation coefficient is required

###### SPECIFIC COMMENTS ######

2. MATERIALS AND METHODS 

Page 4 / Line 148

- Define the capacity of the load cell of the Instron universal testing machine.  Is it 25 kN?

- Define the location where the tests were carried out.

- Considering that the load cell is 25 kN, so the loads being measured are in the order of 2% of the load cell capacity. How do you ensure that there is not a high error in the readings being taken?

- Load cells have a significant error in the early part of their scale. Therefore, they should be used from 10% of the full scale capacity.

Page 5 / Line 179-180

State why the conditions were considered:

"The curing cycle consists of a through thickness and uniform heating at 130 ... , applying a uniform pressure of 0.16 bar."

3. RESULTS AND DISCUSSION

Page 8 /line 274-275

Improve figure 4b) and 4c) . The cubic Fit line is not visualized.

Page 9 /line 314

Figure 5b). How do you ensure that the da/dN curve is representative of the points obtained? Is it not necessary to present the correlation parameter?

- Improve the figures with graphs.

- For all graphs that have theoretical curves, the respective correlation coefficient (Pearson's correlation coefficient) should be presented.

REFERENCES

Page 11

- reference [19] does not exist in the text.

- check that the references are correct.

Author Response

 # In this manuscript, Mode II fatigue delamination growth and healing of Bis-maleimide modified CFRPs were investigated using the MEP technique. The investigation focused on the evaluation of the influence of fatigue life estimation. Mode II fatigue loading conditions were used for delamination growth of CFRPs modified with BMI resin (containing or not containing GNPs). Quasi-static Mode II and Mode II fatigue experimental tests were carried out. The authors found a better delamination behavior after healing activation.

# Interesting subject. Interesting research.

#The research methodology raises some doubts that I would like to see clarified:

The authors would like to thank reviewer #1 for the valuable comments on the manuscript!

  1. How valid are the fatigue tests? I think the load cell capacity is too large for the values read.

Answer:

The fatigue tests are valid. The fatigue tests have been performed under displacement control using the 25 kN UTM. There was no issue with the load measurements since the utilized load cell had a capacity of 1 kN (INSTRON dynamic load cell) and had been connected in series. The 25 kN value represents the nominal characteristics of the used Instron UTM. The relative information has been incorporated into the 2.5 section of the manuscript. Figure 1 illustrates the obtained load cell having a capacity of 1 kN. If the reviewer needs supplementary information about the purchase, we can send it. Thank you a lot!

Figure 1. Illustration of the obtained load cell, having capacity of 1 kN.

  1. the validation of the correlation curves. Pearson's correlation coefficient is required

 Answer:

The Pearson's correlation coefficient (R) has been provided where appropriate.

###### SPECIFIC COMMENTS ######

  1. MATERIALS AND METHODS

Page 4 / Line 148

  1. Define the capacity of the load cell of the Instron universal testing machine.  Is it 25 kN?

Answer:

No, the capacity of the utilized load cell during the tests was 1 kN connected in series with the nominal load cell of the machine. The information has been incorporated into the manuscript.

  1. Define the location where the tests were carried out.

Answer:

The precise location in which the tests were executed has been incorporated into the manuscript (see paragraph 2.5, last sentence).

  1. Considering that the load cell is 25 kN, so the loads being measured are in the order of 2% of the load cell capacity. How do you ensure that there is not a high error in the readings being taken?

Answer:

See answers of 1 and 3 comments.

  1. Load cells have a significant error in the early part of their scale. Therefore, they should be used from 10% of the full-scale capacity.

Answer:

See answers of 1 and 3 comments.

Page 5 / Line 179-180

State why the conditions were considered:

  1. The curing cycle consists of a through thickness and uniform heating at 130 ... , applying a uniform pressure of 0.16 bar."

Answer:

The precise healing profile is the one that has been used in a wide range of publications by the authors for this type of healing agent. The healing profile has been selected after conducting DSC experiments on the selected self-healing agent to initially identify the melting point. Afterwards, an optimization procedure had been conducted considering the temperature, the time and the applied pressure by executing Mode I interlaminar fracture toughness tests. In addition, information arose from the literature for this class of materials has been taken into consideration. Based on that, the selected healing profile was the optimal one. 

  1. RESULTS AND DISCUSSION

Page 8 /line 274-275

  1. Improve figure 4b) and 4c). The cubic Fit line is not visualized.

Answer:

The Figure has been improved in order the fit lines to be more visible.

Page 9 /line 314

  1. Figure 5b). How do you ensure that the da/dN curve is representative of the points obtained? Is it not necessary to present the correlation parameter?

Answer:

The best fitted curve on the obtained experimental points has been incorporated by using the Origin software. The correlation parameter has been provided where appropriate.

  1. Improve the figures with graphs.

Answer:

The figures have been improved where appropriate.

  1. For all graphs that have theoretical curves, the respective correlation coefficient (Pearson's correlation coefficient) should be presented.

Answer:

The Pearson's correlation coefficient has been provided where appropriate. The theoretical curves arose after applying the best fitting curve by using the Origin software.

REFERENCES

Page 11

  1. reference [19] does not exist in the text.

Answer:

The references have been appropriately corrected.

  1. check that the references are correct.

Answer:

The references have been appropriately corrected.

Author Response

The review article, Mode II Fatigue Delamination Growth and Healing of Bis-maleimide Modified CFRPs by Using the Melt Electro-writing Process Technique” describes an essential issue of interlaminar fracture toughness of carbon fiber reinforcement plastic and healing properties as the mechanical performance is critical in many applications.

The article contains a comprehensive description of the sample preparation and testing procedure. However, some deficiencies in the report need to be improved.

The authors would like to thank reviewer #2 for the valuable comments on the manuscript.

  1. First, please carefully revise all the misspellings encountered in the article. Organize carefully as well the figure numbers - Figure 4 appears twice in the manuscript and makes the receipt of work much more challenging.

Answer:

The manuscript has been appropriately revised. The Figure numbers have been carefully corrected. 

  1. Next, the different types of fitting appear in the Figures: polynomial, exponential, cubic – I do not really understand the reason for such fits if no further explanation is given. It should be given why each fit was chosen and the outcomes of using such fits by, e.g. comparing with the literature or citing the previously used equations.

Answer:

The different types of fitting which appear in the Figures, is related to the application of the best fitted curve considering the obtained experimental results by using the Origin software. The reason why for example in Figure 6 appear two different equations (exponential for the reference CFRP while cubic for both Modified CFRPs) is related to the modification procedure that made the modified CFRPs to differently respond. In general, the equations presented within the manuscript are in line with the relative literature in the area (see references 28-29).

  1. The presentation of the results in graphs should also be improved as it looks like preparing in a rush now.

Answer:

The graphs have been appropriately improved where appropriate.

  1. The description of the Figures and their analysis is far from the accepted, e.g. Figure 4 (the second Figure 4) is commented in 6 lines, while contains 4 parts. I do not understand the fit explanation in Figure 4b – it does not align with the results and is not commented on in the text. It needs to be improved, as I see it as the weakest point of this article.

Answer:

The Figure 4 indeed contains four parts but 3 of the 4 parts (Figure 4b-d) contains the same information that is related to the under examination three materials sets (crack propagation curves). The Figure 4b has been improved after removing the outliers. 

  1. I would also advise comparing the healing properties with the literature, as it does not look impressive now. The comparison with the other results of other groups will enhance the receipt of delivered here.

Answer:

A comparison with other investigations has been made within the manuscript.

  1. Also, a deeper analysis of all the results is required as at least 19 (!) of 34 cited articles are the articles of the authors. Please provide a complete comparison of your results with the works of the other authors. I understand your proficiency in the presented field, but at any time a careful literature analysis in reference to the presented results is essential.

Answer:

The self-citation has been reduced from 19 to 7 references as also requested by the editor and more works that are related to other investigations by other experts in the field have been included to the manuscript.

Round 2

Reviewer 1 Report

The authors have made significant improvements to the manuscript. However, there are some points that could be improved. For example:

Answer #2:

In figure 6 Pearson's correlation coefficients are not shown. Why don't you follow the same procedure as in figure 7?

Answer #8:

Page 9 / line 280

Figure 6b) and 6c). The lines are still difficult to see.

To visualize better the curve, change the markers to unfilled, e.g.

Answer #9 and #11:

"The theoretical curves arose after applying the best fitting curve by using the Origin software." OK! but the best fitting curves have a strong correlation or a weak correlation? Do you know?

Answer #9 and #11:

In figure 5, Pearson's correlation coefficients have moderate values. Can't you get a curve with a strong or very strong correlation?

Reviewer 2 Report

The corrections made are satisfactory. I have no further comments.